# Age-Related Alterations in the Behavior and Serotonin-Related Gene mRNA Levels in the Brain of Males and Females of Short-Lived Turquoise Killifish (*Nothobranchius furzeri*)

**DOI:** 10.3390/biom11101421

**Published:** 2021-09-28

**Authors:** Valentina S. Evsiukova, Elizabeth A. Kulikova, Alexander V. Kulikov

**Affiliations:** 1Department of Psychoneuropharmacology, Federal Research Center Institute of Cytology and Genetics, Siberian Branch of the Russian Academy of Sciences, 630090 Novosibirsk, Russia; evsiukova@bionet.nsc.ru (V.S.E.); kulikova@bionet.nsc.ru (E.A.K.); 2Department of Genetic Collections of Neural Disorders, Federal Research Center Institute of Cytology and Genetics, Siberian Branch of the Russian Academy of Sciences, 630090 Novosibirsk, Russia

**Keywords:** turquoise killifish, aging, sexual dimorphism, novel tank diving test, tryptophan hydroxylase, monoamine oxidase, serotonin transporter, serotonin receptors, gene expression, brain

## Abstract

Short-lived turquoise killifish (*Nothobranchius furzeri*) have become a popular model organism for neuroscience. In the present paper we study for the first time their behavior in the novel tank diving test and the levels of mRNA of various 5-HT-related genes in brains of 2-, 4- and 6-month-old males and females of *N. furzeri*. The marked effect of age on body mass, locomotor activity and the mRNA level of *Tph1b*, *Tph2*, *Slc6a4b*, *Mao*, *Htr1aa*, *Htr2a*, *Htr3a*, *Htr3b*, *Htr4*, *Htr6* genes in the brains of *N. furzeri* males was shown. Locomotor activity and expression of the Mao gene increased, while expression of *Tph1b*, *Tph2*, *Slc6a4b*, *Htr1aa*, *Htr2a*, *Htr3a*, *Htr3b*, *Htr4*, *Htr6* genes decreased in 6-month-old killifish. Significant effects of sex on body mass as well as on mRNA level of *Tph1a*, *Tph1b*, *Tph2*, *Slc6a4b*, *Htr1aa*, *5-HT2a*, *Htr3a*, *Htr3b*, *Htr4*, and *Htr6* genes were revealed: in general both the body mass and the expression of these genes were higher in males. *N. furzeri* is a suitable model with which to study the fundamental problems of age-related alterations in various mRNA levels related with the brains 5-HT system.

## 1. Introduction

The brain’s serotonin (5-HT) system plays a key role in the regulation of neuronal plasticity [1], numerous physiological functions and various kinds of behavior [2]. Its dysfunctions are associated with psychopathologies such as depression, anxiety, obsessive-compulsive syndrome, etc [3,4,5,6,7].

Age-related alterations in the brain’s 5-HT system frequently accompany psychic disorders in senior patients [8]. The majority of knowledge concerning age-related changes in the brain’s 5-HT system was based on results obtained from aged people (60 years old or older) using the PET assay and post mortem studies as well as experiments carried out on laboratory monkeys and rodents (see reviews) [9,10,11]. However, these studies are dramatically limited due to ethical norms and the relatively long lifespan of laboratory monkeys (>20 years old) and rodents (2 years old).

The short-lived turquoise killifish, *Nothobranchius furzeri*, inhabits ephemeral ponds in southeastern Africa. It’s fertilized eggs survive the dry season in diapause. The larvae hatch immediately after the pond is filled with water, grow rapidly, reach sexual maturity within four to six weeks, continuously mate and spawn during the wet season, and die at the age of six to eight months [12,13,14]. Six- to eight-month-old *N. furzeri* show morphological and behavioral hallmarks of aging, such as reduced coloration in males, emaciation, spinal curvature, spine and face malformations [12,14], reduced spontaneous, exploratory activities [15] and impaired learning performance [16,17].

The brain’s 5-HT system regulates the physiological functions and behavior of killifish. Indeed, chronic treatment with small doses of the 5-HT transporter blocker, fluoxetine, led to decreased body size, as well as increased fecundity and sociability in *N. furzeri* [18,19]. However, until now there was only one publication revealing no effect of sex and rearing conditions on the level and turnover of 5-HT in brains of 108-day-old *N. furzeri* [20]. The fish brain’s 5-HT system is homologous to that of mammals [21]. The fast life-cycle, hallmarks of aging and possible homology of *N. furzeri* make this species a convenient model to elucidate the alterations in the brain’s 5-HT system and 5-HT-related functions in aging.

The novel tank diving test is the most frequently used test to demonstrate the conflict between exploration and anxiety in zebrafish (*Danio rerio*) [22,23]. 5-HT seems to regulate zebrafish’s reaction to novelty [24].

The aim of the present study was to investigate age-dependent alterations in the 5-HT dependent behavior and mRNA levels of the 5-HT-related genes encoding the key enzymes of metabolism, transporter and receptors of 5-HT in the brain of *N. furzeri*. This approach provides complex information about the alterations in the brain’s 5-HT system activity in killifish during aging. Here we intended to compare the locomotor activity and anxiety-related behavior in the novel tank diving test as well as mRNA levels of all sequenced 5-HT-related genes such as *Tph1a*, *Tph1b*, *Tph2*, *Slc6a4b*, *Mao*, *Htr1aa*, *Htr1b*, *Htr1f*, *Htr2a*, *Htr2cl1*, *Htr3a*, *Htr3b*, *Htr4*, *Htr5ab*, *Htr6*, and *Htr7* in brains of 2-, 4- and 6-month-old males and females of *N. furzeri*.

## 2. Materials and Methods

### 2.1. Animals

The experiments were carried out on 37 males and 27 females of *N. furzeri* of the ZMZ1001 strain. The progenitors of these killifish were received from the European Research Institute of Biology of Ageing (ERIBA, Groningen, Netherlands) [20]. The breeding and hatching of these fish were carried out in the fish facility of the Institute of Cytology and Genetics SB RAS (Novosibirsk, Russia) according to the published protocol [25]. The larvae were fed *ad libitum* three times per day with freshly hatched brine shrimps (*Artemia salina*) for the first three weeks after hatching. At the age of 21 days, post-hatch young fish were reared in mixed-sex groups in 125 L glass tanks (75 cm in length, 40 cm in depth, 48 cm in width, water depth was 35 cm). The sizes of these groups changed from 30 at the beginning to 15 at the end of experiment. Water in the tanks was constantly filtered and aerated with a filter, a model XL-860 (Xilong, Zhongshan, China), and its temperature was 27 ± 1 °C. Tanks were equipped with three plastic plants 14 cm in height each and two dishes (10 cm in diameter, 3 cm in height) filled with sand for spawning. The plastic plants served as environmental enrichment and as shelters from aggressive conspecifics. Every day (at 17:00) the tanks were cleaned and 10% of the water was substituted with tap water filtered through the expert hard filter Barrier Expert Standard (BVT Barrier Rus, Noginsk, Russia). The 12 h light/12 h dark (“light on” mode at 09:00) photoperiod was maintained. Young and adult killifish were fed until satiation two times per day with frozen blood worms (Chironomus plumosus). To avoid competition for food, we added as many worms as the fish could eat in an hour. The fish were kept in these tanks until the novel tank diving test.

There were three age groups of killifish: 60 days old (13 males and 8 females), 120 days old (12 males and 11 females) and 180 days old (12 males and 8 females). Each group consisted of males and females caught randomly from the tanks.

### 2.2. Novel Tank Diving Test

When we started carrying out the test, the age of males and females was 60, 120 and 180 days. The test was conducted in the daytime (11:00 a.m.–3:00 p.m.) in a glass test tank (24 cm in length, 15 cm in depth and 7 cm in width) according to the published protocol [20]. The fish were individually transferred from their home tank into the test one and the recording started immediately. The water column (24 cm in length and 10 cm in depth) was virtually divided into the lower, middle and upper thirds (24 cm in length and 3.3 cm in height each). The test duration was 5 min. The fish position was automatically recorded for 5 min at rate of 30 fps by a C920 Pro HD Web camera (Logitech, Lausanne, Switzerland) connected to a computer (Windows 7) via a USB 2.0 port. The stream of frames was automatically analyzed in real-time by the EthoStudio software (Institute of Automation and Electrometry, Siberian Branch of Russian Academy of Sciences, Novosibirsk, Russia) [26] and saved on a hard disk as a compressed video file. The EthoStudio software separated the pixels associated with the fish from those associated with the background frame by frame, applying the threshold algorithm, [26] and calculated the coordinates of the fish center. Moreover, the EthoStudio software automatically calculated the density map corresponding to the spatial distribution of fish-associated pixels in the tank [26,27]. The sequence of coordinates and the density map were used to evaluate (1) the distance travelled (cm); (2) the immobility time (%); (3) the mean distance from the tank’s bottom (cm); the time spent (%) in (4) the lower and (5) the upper thirds of the tank,; and(6) the explored part of the tank (%). The fourth and fifth parameters were calculated as the ratios of the amount of fish-associated pixels in the lower and upper thirds of the tank, respectively, to the sum of fish-associated pixels in all three thirds [18,26,27]. The last parameter was calculated as the ratio of the amount of pixels visited by the fish to the total amount of pixels in the tank’s water area [28]. Each fish was recorded only once.

### 2.3. qPCR

Immediately after the novel tank diving test the fish were euthanized by immersion into 0.1% tricaine methanesulfonate (Sigma-Aldrich, St. Louis, MO, USA) solution and then into cold water (+2 C), their bodies were immediately dried with dry napkins and their masses were measured using an Ohaus PA-512 electronic balance (Ohaus Corporation, Parsippany, USA) with an accuracy of 10 mg. Then their whole brains were immediately removed, frozen with liquid nitrogen and stored at −80 °C [20]. Because of the small size of the brain, the method sensitivity and fish numbers, we could reliably quantify the gene expression only in the whole brain. The brain was homogenized in 300 µL of Trizol reagent (Bio Rad, Hercules, CA, USA) using a motor-driven grinder (Z359971, Sigma-Aldrich, St. Louis, MO, USA). Total mRNA extraction; treatment of RNA with RNAase free DNAase (Promega, Madison, WI, USA); cDNA synthesis with a random hexanucleotide primer and R01 Kit (Biolabmix, Novosibirsk, Russia) and SYBR Green real-time quantitative PCR with selective primers (Table 1) and R401 Kit (Sintol, Moscow, Russia) were performed according to the manufacturers’ protocols. RNA quality was checked by electrophoresis in 1% agarose. Only the RNA samples with two clear bands of ribosomal RNA were used for the qPCR study. The primer pairs’ design was based on the published sequences of the target genes (https://www.ensembl.org/index.html (accessed on 30 August 2021)). As for the external standards, we used solutions containing 50, 100, 200, 400, 800, 1600, 3200, 6400 and 12,800 copies of genomic DNA extracted from killifish muscle. The gene expression was presented as a relative number of cDNA copies calculated on 100 copies of *Polr2eb* cDNA as internal standard [29,30].

### 2.4. Statistics

All data were tested using Kolmogorov’s test and met the assumption of normality. Data were presented as the mean ± SEM and analyzed by two-way ANOVA with “Age” and “Sex”, including their interaction, as the independent factors. *Post hoc* analyses were carried out using the Fisher’s LSD multiple comparison test when appropriate. In addition, the number of variables was reduced through principal component analysis with factor varimax normalized rotation using Statistics 8.0 (StatSoft, Inc, Tulsa, OK, USA). For principal component analysis, the values of immobility time, explored part of the tank, time spent in the lower and upper thirds of the tank were transformed by square root extraction. Then the factors were analyzed through discriminant analysis. Statistical significance was set at *p* < 0.05. The significance of correlation coefficients was corrected according to Bonferroni.

## 3. Results

### 3.1. Body Mass of 2, 4 and 6-Month-Old Males and Females of N. furzeri

Marked effects of “Age” (F_2,57_ = 8.73, *p* = 0.0005), and“Sex” (F_1,57_ = 72.09, *p* < 0.0001) factors, but not their interaction (F_2,57_ < 1) on body mass, were revealed. Six-month-old males were heavier than two-month-old ones (Figure 1). However, no difference in body mass between two, four and six-month-old females was shown (Figure 1). Males were heavier than females of the same age (Figure 1).

### 3.2. Behavior in the Novel Tank Diving Test of Two-, Four- and Six-Month-Old Males and Females of N. furzeri

A significant effect of the “Age” factor on distance travelled and mean distance from the tank’s bottom was revealed (Table 2). We also observed as a tendency the effect of the “Age” factor on immobility time (*p* = 0.052, Table 2). Four- and six-month-old males did not differ from two-month-old males in distance travelled, mean distance from the tank’s bottom and immobility time. At the same time, the distance travelled was higher, while immobility time and mean distance from the tank’s bottom were lower in six-month-old fish compared to four-month-old ones (Figure 2). No age-dependent alteration in the explored part of the tank or time spent in the lower and upper thirds of the tank was observed in males (Figure 2). Females of all studied age groups did not differ in these traits (Figure 2).

A marked effect of the “Sex” factor on immobility time was shown (Table 2). This trait decreased in four-month-old females compared to males of the same age (Figure 2). Males and females of all age groups did not differ in distance travelled, explored part of the tank, mean distance from the tank’s bottom, or in time spent in the lower and upper thirds of the tank (Table 2, Figure 2).

Principal component analysis revealed that 92.7% of variability of these six behavioral traits in males could be described by two factors. The first factor correlated with the anxiety-related traits, such as time spent in the lower, upper thirds and mean distance from the tank’s bottom. We named it “anxiety”. The second factor correlated with locomotion traits such as travelled distance, immobility time and explored part of the tank. We named it “locomotion” (Table 3). The two-factors model revealed significant effect of age on behavior in males (F_4,60_ = 3.13, *p* = 0.021). Discriminant analysis revealed differences in behavior between four- and six-month-old males (*p* = 0.0034) (Figure 3). No age-dependent alteration in the first factor was observed (F_2,30_ = 2.66, *p* = 0.0.087). At the same time, significant age-dependent changes of the second factor were witnessed (F_2,30_ = 4.27, *p* = 0.023).

### 3.3. Levels of mRNA of Polr2eb, Tph1a, Tph1b, Tph2, Slc6a4b, Mao, Htr1aa, Htr1b, Htr1f, Htr2a, Htr2cl1, Htr3a, Htr3b, Htr4, Htr5ab, Htr6, and Htr7 Genes in Brain of 2-, 4-, and 6-Month-Old Males and Females of N. furzeri

No effect of “Age”or “Sex” factors and their interaction on the mRNA level of *Polr2eb* gene in the brain of killifish was observed (Table 4). Therefore, we could correctly use this gene mRNA level as an internal standard [27,28].

Marked effect of the “Age” and “Sex” factors interaction on the mRNA level of *Tph1b*, *Slc6a4b*, *Htr1aa*, *Htr2a*, *Htr3a*, *Htr3b*, and *Htr4* genes was shown (Table 4). In addition, a significant effect of the “Age” factor on the mRNA level of *Tph1b*, *Slc6a4b*, *Mao*, *Htr1aa*, *Htr3a*, *Htr3b*, *Htr6* genes was revealed (Table 4). There were four types of age-dependent alterations in gene expression: (1) no statistically significant alteration (*Tph1a*, *Htr1b*, *Htr1f*, *Htr2cl1*, *Htr5ab*, *Htr7)*; (2) age-dependent increase (*Mao*); (3) age-dependent decrease (*Htr1aa*, *Htr4*); and (4) increase at the age of four months followed by a decrease at the age of six months (*Tph1b*, *Tph2*, *Slc6a4b*, *Htr2a*, *Htr3a*, *Htr3b*, *Htr6)* (Table 5).

The levels of mRNA of *Mao* and *Htr1aa* genes gradually increased and decreased, respectively, with age in both males and females (Table 5). No difference in the mRNA level of *Htr4* gene in two- and four-month-old males was observed, but this trait decreased in six-month-old males. At the same time, no age-dependent difference in the *Htr4* gene mRNA level in females was found (Table 5).

The levels of mRNA of *Tph1b*, *Tph2*, *Slc6a4b*, *Htr2a*, *Htr3b*, *Htr6* genes increased in four-month-old males compared to two- and six-months-old ones. However, no age-dependent alteration in these genes’ mRNA levels was observed in the brain of females (Table 5).

No sexual dimorphism for mRNA levels of *Mao*, *Htr1b*, *Htr1f*, *Htr2cl1*, *Htr5ab*, or *Htr7* genes was observed. At the same time, a significant effect of the “Sex” factor on the mRNA level of *Tph1a*, *Tph1b*, *Tph2*, *Slc6a4b*, *Htr1aa*, *5-HT2a*, *Htr3a*, *Htr3b*, *Htr4*, and *Htr6* genes was revealed (Table 4). The sexual dimorphism was dependent on the gene and age. *Tph1a* gene mRNA concentration in males was higher compared to females for all age groups. *Tph2* and *Htr3a* genes mRNA levels in the four- and six-months-old males was higher compared to females of the same age. The mRNA levels of *Tph1b*, *Slc6a4b* and the *Htr3b* genes were higher in two- and four-month-old males than in females of the same age. A significant decrease in *Htr1aa*, *Htr4* and *Htr6* genes mRNA concentrations was observed in four-month-old females compared to males of the same age. At the same time, the mRNA level of the *Htr2a* gene was higher in the brain of two- and six-month-old females compared to males of the same age. No age-dependent dynamics in *Tph1b*, *Tph2*, *Slc6a4b*, *Htr1aa*, *5-HT2a*, *Htr3a*, *Htr3b*, *Htr4*, or the *Htr6* genes expression in female brains was shown.

Principal component analysis revealed that 82.5% variability in the levels of 15 transcripts in the brains of males could be described by three factors. The first factor correlated with *Tph1b*, *Tph2*, *Slc6a4b*, *Htr1aa*, *Htr1b*, *Htr1f*, *Htr2cl1*, *Htr3a*, *Htr3b*, *Htr4*, *Htr5ab and Htr6* genes expression. It reflects the decrease in these genes’ expression in 6-month-old males and we named it age-related decrease (Table 6, Figure 4). The second one correlates with *Mao* and *Htr7* genes expressions. It reflects increase in these genes’ expression in six-month-old males and we named it “age-related increase” (Table 6, Figure 4). The third factors correlated with the *Tph1a* gene expression (Table 6). The three-factors model revealed a significant effect of age on behavior in males (F_6,58_ = 7.44, *p* < 0.0001). Discriminant analysis revealed a difference in the genes expression between two- and four- (*p* = 0.0067), two- and six- (*p* = 0.0005), four- and six- (*p* = 0.0001) month-old males (Figure 4). Age-dependent alterations in the first (F_3,29_ = 10.85, *p* = 0.0003), second (F_3,29_ = 8.57, *p* = 0.0012) and third (F_3,29_ = 3.66, *p* = 0.038) factors were revealed.

### 3.4. Correlation between Behavioral Traits in Novel Tank Diving Test and mRNA Levels of 5-HT-Related Genes in the Brain of Two-, Four- and Six-Month-Old Males of N. furzeri

No statistically significant correlation between the travelled distance, immobility time, explored part of the tank, distance from the tank’s bottom, time spent in the lower and upper thirds in the novel tank diving test on the one hand and the mRNA levels of 16 5-HT related transcripts in the brains on the other hand in two-, four- and six-month-old males was found (Table 7).

## 4. Discussion

In this study we compared the behavior in a novel tank diving test and the levels of mRNA of 5-HT related genes in the brain in three age groups of killifish: (1) relatively young (two months old); (2) middle aged (four months old); and (3) old (6sixmonths old) when natural death of fish begins to occur [12,13,14]. First of all, we found that males were heavier than females of the same age. This result confirms earlier obtained data [15,18,20,31,32]. Body masses of males progressively increased with age and old males were significantly heavier than young ones. However, age-dependent increase in body masses in females was less intensive than those of males. Thus, we did not observe any difference between young, middle aged and old females.

Turquoise killifish is a promising model species for neuroscience, psychopharmacology and ecotoxicology [18,19,31,32,33,34,35]. Behavioral variation in males and females of killifish was shown in the emergence, open field, habitat choice, life skill tests [33] and diurnal activity [34]. However, no sexual dimorphism in killifish behavior in open field, habitat choice and life skill tests was revealed [33]. Earlier we did not find difference in locomotor- and anxiety-related traits between males and females of *N. furzeri* [20]. However, in the present study we reveled an increase of immobility time in four-month-old males compared to females of the same age.

A marked decline in spontaneous, exploratory activities [15] and impaired learning performance [16,17] in old N. furzeri were shown. In the present study, significant age-related alterations in distance travelled and immobility time in males, but not in females were shown. Males of the middle aged group (four-months-old) travelled less distance and were more immobile compared to males from the old group. Principal component analysis of six behavioral parameters in males confirmed this conclusion and showed age-related variability in the factor associated with locomotor activity. The observed increase in locomotor activity in old males compared to middle aged males seems paradoxical since decrease in locomotor activity in aged *N. furzeri* is well established [15]. We think that the main cause of this paradoxical increase of locomotion is the reduction of the freezing reaction to novelty in old compared to middle aged males: immobility, an index of freezing, in old males was lower than in middle aged males.

Currently, the key role of the brain’s 5-HT system in the regulation of behavior in the novel tank diving test in zebrafish is beyond doubt [24]. Indeed, 5-HT_1A_ receptor agonists decrease the time spent near the tank’s bottom [36] and 5-HT reuptake blockers cause so called “surface dwelling” [27,36,37,38]. The brain’s 5-HT system also regulates the physiological functions and behavior of killifish: chronic treatment with small doses of 5-HT the transporter inhibitor fluoxetine decreased body size as well as increased fecundity and sociability in *N. furzeri* [18,19]. Moreover, chronic fluoxetine administration reduced travelled distance and total moving time in killifish [34].

The brain’s 5-HT system is highly conservative in all vertebrates from fish to mammals [39]. The bodies of 5-HT neurons are mapped in the midbrain while their endings are found in all brain regions [34]. The key enzymes of 5-HT synthesis in mammals are tryptophan hydroxylase 2 (TPH2) in neurons and tryptophan hydroxylase 1 (TPH1) in peripheral tissues those hydroxylate essential amino acid L-tryptophan to 5-hydroxytryptophan [39,40]. Unlike mammals, in the genome of *N. furzeri Tph2*, *Tph1a* and *Tph1b* genes are found (Table 8). The synthesized 5-HT is stored in vesicles, transported to 5-HT endings, secreted into the synaptic cleft and then it interacts with numerous 5-HT receptors. Fourteen different subtypes of 5-HT receptors coupled to G_i_ (5-HT1A, 5-HT1B, 5-HT1D, 5-HT1E, 5-HT1F, 5-HT5A and 5-HT5B), G_q_ (5-HT2A, 5-HT2B, 5-HT2C), G_s_ (5-HT4, 5-HT6, 5-HT7) proteins or Na^+^ ion-gated channels (5-HT3) are found in the brain of mammals [41,42]. At the same time, in the genome of *N. furzeri*, only 10 genes (*Htr1aa*, *Htr1b*, *Htr1f*, *Htr5ab*, *Htr2a*, *Htr2cl1*, *Htr3*, *Htr4*, *Htr6*, and *Ht7*) encoding 5-HT receptors were found (Table 8). Protein 5-HT transporter (SERT) reuptakes 5-HT from the synaptic cleft into 5-HT neurons [43] where it is restored in vesicles or is oxidized to 5-hydroxyindole acetic acid by monoamine oxidase A and B enzymes (MAOA, MAOB) [44,45]. In the genome of *N. furzeri Scl6a4b* gene encodes SERT and the only *Mao* gene encodes the only enzyme MAO (Table 8).

Here we found that *Tph1a* and *Tph1b* mRNA levels in brain of males and females of *N. furzeri* are even higher than *Tph2*. It should be mentioned that data concerning TPH1 expression in the brain are rather contradictory: some authors reported an extremely low *Tph1* mRNA level in rat brains [46,47], while other authors found relatively high *Tph1* gene expression in rat [48,49] and human [50,51,52] brains.

In this research we investigated the effects of sex and age on fifteen 5-HT related genes in the brains of *N. furzeri* for the first time.

First of all, we showed marked sexual dimorphism in *Tph1a*, *Tph1b*, *Tph2*, *Slc6a4b*, *Htr1aa*, *Htr3a*, *Htr3b*, *Htr4*, and *Htr6* genes expression. The mRNA levels of these genes are higher in middle aged males compared to females of the same age. The cause of this dimorphism is unknown. Moreover, middle aged males and females did not differ in the levels of 5-HT and 5-HIAA in their brains [20].

It should be noted that in males, age-related alterations were shown for expression of 10 genes, *Tph1b*, *Tph2*, *Mao*, *Slc6a4b*, *Htr1aa*, *Htr2a*, *Htr3a*, *Htr3b*, *Htr4*, and *Htr6*, while in females -this only occurred only for two genes, *Mao* and *Htr1aa*.

We showed biphasic dynamics of the *Tph2* mRNA level in brains of killifish males: it rose in middle aged males and then again fell in old fish. Age-dependent alterations of TPH2 in the mammalian brain are obscure. Some authors did not find any difference in the Tph2 mRNA level in the brain structures between young and old rats [53]. However, other authors reported increases and decreases of TPH2 activity in the midbrain and medulla, respectively, of old rats compared to middle aged ones [54].

A progressive elevation of the Mao transcript level in brains of middle aged and old males and females compared to young killifish was revealed. This result agrees well with elevation of the MAOA activity in the brains of rats [55] and humans [56,57,58] during aging.

We showed a biphasic dynamic of the *Slc6a4b* mRNA level in brains of killifish males: it rose in middle aged males and then again fell in older animals. This dynamic agrees with the drop of the SERT protein level in old mice [59], rats [60], hamsters [61] and humans [62].

If TPH2, MAO and SERT seem to play similar functions in killifish and mammals; there is no knowledge about functions of 5-HT receptors in killifish. Here we studied the levels of mRNA of 10 sequenced 5-HT receptor-like genes in the killifish brain. These genes can be conditionally divided into three groups, with high (*Htr1aa*, *Htr1f*, *Htr2a*, *Htr2cl1*), medium (*Htr1b*, *Htr4*, *Htr5ab*, *Htr7*) and low (*Htr3a*, *Htr3b*, *Htr6*) expression in the killifish’s brain. This distribution of different types of 5-HT receptors in killifish brain is in good concordance with that of the mammalian brain [41].

The level of mRNA of *Htr1aa* gene encoding the 5-HT1A receptor progressively decreased in brains of old males and females of *N. furzeri*. Some authors also reported the reduction of 5-HT1A binding sites in brains of old rats [63], hamsters [61], monkeys [64] and humans [65,66]. However, other investigators did not find any difference in the *Htr1a* gene mRNA level in brains of young and old rats [53,67].

*Htr2a* gene expression in the brain of old males was lower than in middle aged males. Some authors also reported age-dependent reduction of 5-HT2A receptor binding sites in brains of old rats [68], monkeys [64] and humans [66,69]. However, other authors did not find any difference in the *Htr2a* gene mRNA in brains of young and old rats [67].

The age-dependent decline in the mRNA levels of *Htr4* and *Htr6* genes in the brain of *N. furzeri* males shown in our study is in good concordance with that in the 5-HT4 [70] and 5-HT6 [69,71] binding sites in brains of aged humans.

The absence of age-dependent dynamics of *Htr7* mRNA concentration in the brains of *N. furzeri* agrees with that in brains of hamsters [72] and rats [67].

We did not reveal age-related alterations in the level of *Htr1b* gene mRNA in brains of N. males and females. However, other investigators showed decline in 5-HT1B binding sites [69] and the *Htr1b* mRNA level [53] in brains of old rats compared to young animals.

We did not find published information concerning the age-dependent dynamics of 5-HT1F, 5-HT5 and 5-HT3 genes or protein expression in vertebrates. Here, we did not observe any sex and age-dependent alterations in the levels of the *Htr1f* and *Htr5ab* gene mRNA in killifish brains. At the same time, the brain levels of *Htr3a* and *Htr3b* mRNA significantly increased in four-month-old males and then again decreased in six-month-old fish.

Principal component analysis revealed three patterns (factors) of age-dependent alteration in the gene expression in killifish males. The first pattern reflects an age-dependent decline of *Tph1b*, *Tph2*, *Slc6a4b*, *Htr1aa*, *Htr3a*, *Htr3b*, *Htr4*, and *Htr6* gene expression. The second pattern reflects an age-dependent elevation of mRNA levels of *Mao* gene expression. Since MAO is linked to the mitochondrial membrane, this factor can be associated with the age-dependent dysfunction of these organelles. The third pattern correlates with *Tph1a* gene expression. A biological sense of this factor is still unknown, since there is no information about the role that TPH1A plays in brain of *N. furzeri*.

Although there is pharmacological evidence that 5-HT regulates locomotor activity in *N. furzeri* [18,19,34], in the present study we failed to show any statistically significant correlation between the expression of sixteen 5-HT-related transcripts and the locomotor activity in the novel diving test in killifish males. However, the age-dependent alterations in locomotion could be correlated with those of 5-HT related gene expression in the brain regions involved in the regulation of locomotor activity.

## 5. Conclusions

This is the first study on the effects of sex and age on the 5-HT system and 5-HT-dependent behavior in the brain in the novel tank diving test in short-lived turquoise killifish (*N. furzeri*).

We found the following effect of aging on locomotion in the novel tank diving test in *N. furzeri* males: immobility (freezing) as a reaction to novelty increased in middle aged males and then again decreased in old males, while females did not show any age-related alterations in locomotion in this test.

Selective probes for qPCR quantification were developed and the expression of *Tph1a*, *Tph1b*, *Tph2*, *Slc6a4b*, *Mao*, *Htr1aa*, *Htr1b*, *Htr1f*, *Htr2a*, *Htr2cl1*, *Htr3a*, *Htr3b*, *Htr4*, *Htr5ab*, *Htr6*, and *Htr7* in the brains of young (two-month-old), middle aged (four-month-old) and old (six-month old) males and females of *N. furzeri* were studied.

Despite the fact that, due to methodical limitations, we could measure the expression of 5-HT related genes only in the whole brain, we revealed significant sexual and age-related changes in the expression of some of these genes.

Remarkable sexual dimorphism in *Tph1a*, *Tph1b*, *Tph2*, *Slc6a4b*, *Htr1aa*, *Htr3a*, *Htr3b*, *Htr4*, and *Htr6* gene expression was shown: their expressions in males were higher than in females.

Two important patterns of age-dependent alterations in these genes expression in killifish male brain was revealed: the mRNA levels of *Tph1b*, *Tph2*, *Slc6a4b*, *Htr1aa*, *Htr2a*, *Htr3a*, *Htr3b*, *Htr4 and Htr6* genes decreased, while *Mao* gene mRNA levels increased in old males. At the same time, age-related alterations in gene expression in the brain of *N. furzeri* females were more modest: we revealed only elevation of the *Mao* gene mRNA level and reduction of *Htr1aa* gene expression in old females.

These age-related changes of these 5-HT related gene expression in the brain of *N. furzeri* males demonstrate a marvelous similarity with aging alterations in TPH2 and MAOA activities as well as SERT and 5-HT receptor binding sites in the brains of rats, monkeys and humans (Table 9). Thus, the observed age-dependent alterations in these 5-HT related gene alterations in brains of *N. furzeri* indeed reflect the fundamental changes in the vertebrate 5-HT system during aging.

It should be highlighted that six-month-old *N. furzeri* correspond to 24-month-old rats and mice. Here we have shown that the age-dependent alterations in the brain’s 5-HT system in *N. furzeri* can model the aging changes in the brains of laboratory rodents, monkeys and even humans. Therefore, *N. furzeri* is a promising model species that can significantly accelerate the study of aging effects on the brain’s 5-HT system.

## Figures and Tables

**Figure 1 biomolecules-11-01421-f001:**
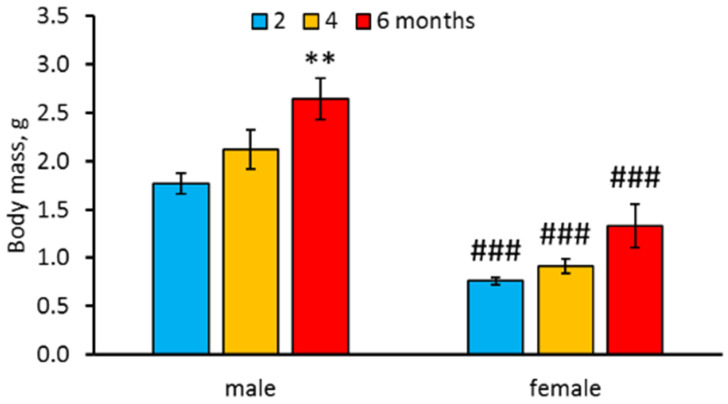
Body mass of two-, four-, and six-month old males and females *N. furzeri*. ** *p* < 0.01 vs. two-month-old males, ^###^
*p* < 0.001 vs. males of the same age.

**Figure 2 biomolecules-11-01421-f002:**
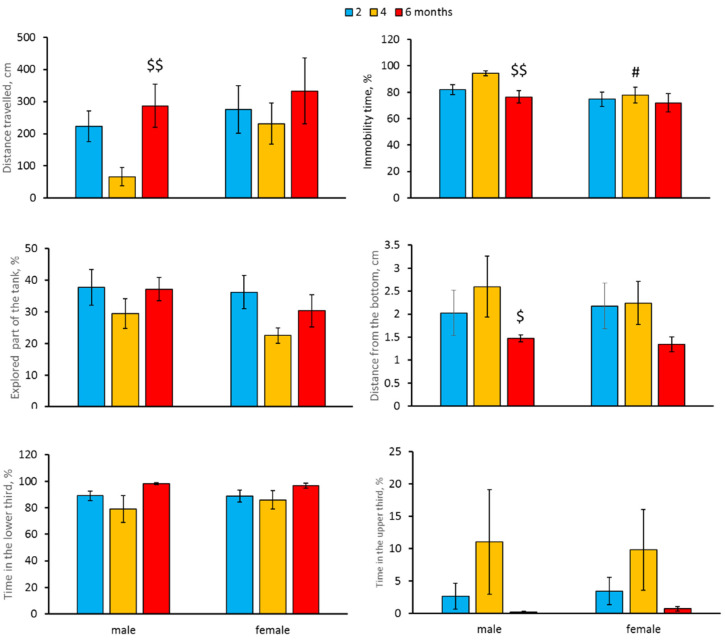
Distance travelled (cm), immobility time (%), explored part of the tank (%), mean distance from the tank’s bottom (cm), time spent (%) in the lower and upper third of the tank in the novel tank diving test in two-, four- and six-month-old males and females *N. furzeri*. ^$^
*p* < 0.05, ^$$^
*p* < 0.01 vs. four-month-old males, ^#^
*p* < 0.05 vs. males of the same age.

**Figure 3 biomolecules-11-01421-f003:**
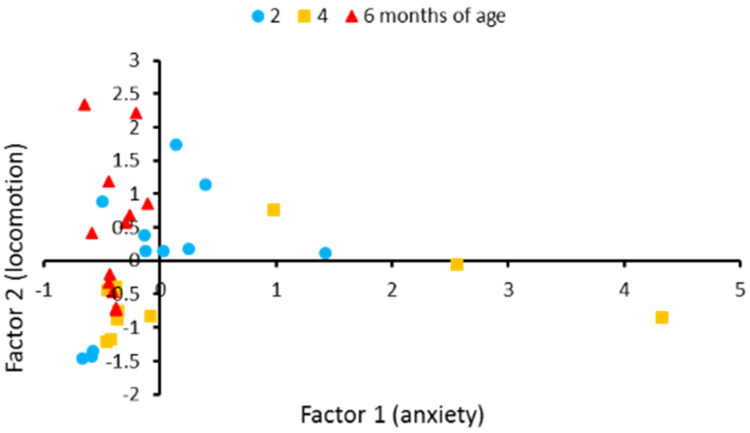
Individual scores for eleven two-month-old, eleven four-month-old, and twelve six-month-old males of *N. furzeri* along two factors yielded by the principal component analysis. These factors generalize six behavioral traits in the novel tank diving test. Factor 1 loads the anxiety-related traits such as distance from the tank’s bottom and time spent in the lower and upper thirds. Factor 2 loads the traits corresponding to locomotor and exploratory activities such as travelled distance, immobility time and the explored part of the tank. The coordinates of the two axes represent the factor scores of individual animals. Ten of 11 scores corresponding to four-month-old males are located in the lower half of the graphic, while all 12 scores corresponding to six-month-old males are located in the left half of graphic.

**Figure 4 biomolecules-11-01421-f004:**
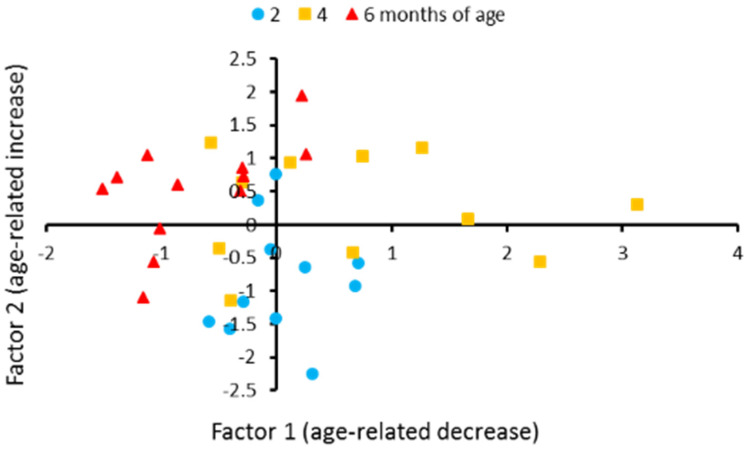
Individual scores for eleven two-month-old, eleven four-month-old and twelve six-month-old males of *N. furzeri* along with factor 1 and factor 2 yielded by the principal component analysis. These factors generalize mRNA levels of 14 of 16 5-HT related genes in the brain. Factor 1 loads the genes those expression decreased in old males, such as *Tph1a*, *Tph2*, *Slc6a4b*, *Htr1aa*, *Htr3a*, *Htr3b*, *Htr4* and *Htr6*. Factor 2 loads *Mao* gene that expression increased in old males. The coordinates of the two axes represent the factor scores of individual animals. Ten of 12 scores corresponding to 6six-month-old males are located in the left half of the graphic. Nine of 11 scores corresponding to two-month-old males are located in the lower half of the graphic.

**Table 1 biomolecules-11-01421-t001:** Sequences and annealing temperatures of the primers.

Gene	Primer Sequences	Annealing Temperatures, °C
*Polr2e*	F 5′-ctgtgccatgatcgaggttac-3′ R 5′-gacgaccttcgctcggttt-3′	64
*Tph1a*	F 5′-agtttgccaagaccatcaag-3′ R 5′-cttgttgagccgattgagag-3′	60
*Tph1b*	F 5′-agcactcaggtttcagcattc-3′ R 5′-ggctcaggcgtgtaaagagg-3′	63
*Tph2*	F 5′-acgaacgtctttcagtcccatc-3′ R 5′-tggaagagtttgagagctttgacc-3′	64
*Mao*	F 5′-ttctcagtaacatgactcaacacttg-3′ R 5′-ttgcttctgtgaactcactgtag-3′	63
*Slc6a4b*	F 5′-agacggattccagggacaag-3′ R 5′-cgtttccaagatccactgcg-3′	63
*Htr1aa*	F 5′-ggaactttcatcgtgtgctgg-3′ R 5′-gatgggattcaggagagagttgg-3′	64
*Htr1b*	F 5′-tggtgtccatcctggtaatgc-3′ R 5′-cagtaccgatccaaagcgattac-3′	64
*Htr1f*	F 5′-atggatttcttcaactggactgaag-3′ R 5′-cagggagcagatgaggtagtt-3′	63
*Htr2a*	F 5′-cctttgttgccttctttgttcc-3′ R 5′-gacaccgaggtttgtggtag-3′	62
*Htr2cl1*	F 5′-tggaccttaacctaacccagc-3′ R 5′-caggaagaagttggtggcatt-3′	63
*Htr3a*	F 5′-ttgtacgtctggtccacaagc-3′ R 5′-tgttttccttgtaatgctccaggt-3′	64
*Htr3b*	F 5′-gttctttgtggtgtgcatggc-3′ R 5′-gatcgagggttttgatggaggtt-3′	64
*Htr4*	F 5′-ccaactatttcatcgtgtccctg-3′ R 5′-gtttcaccgtagatccagttctc-3′	63
*Htr5ab*	F 5′-caggtctggatctcatttgacg-3′ R 5′-ctagtcttgagcgtgtactcc-3′	62
*Htr6*	F 5′-ctagaccgttacctcttcatcatc-3′ R 5′- tcaatgggaaggaaagaagcc-3′	62
*Htr7*	F 5′-gtactacaagatattcagggcagc-3′ R 5′-ggagatgtttcttcgctcgc-3′	63

**Table 2 biomolecules-11-01421-t002:** Two-way ANOVA of the effect of “Age”, “Sex” factors and their interaction on the variability of behavior of *N. furzeri* in the novel tank diving test.

Trait	Age	Sex	Interaction
Distance traveled, cm	**F_2,58_ = 3.39, *p* = 0.04**	F_1,58_ = 2.89, *p* = 0.09	F_2,58_ < 1
Immobility time, %	F_2,58_ = 3.12, *p* = 0.052	**F_1,58_ = 5.45, *p* = 0.02**	F_2,58_ < 1
Explored part, %	F_2,58_ = 3.0, *p* = 0.06	F_1,58_ = 1.74, *p* = 0.19	F_2,58_ < 1
Distance from the bottom, cm	**F_2,58_ = 3.22, *p* = 0.047**	F_1,58_ < 1	F_2,58_ < 1
Time in the lower third, %	F_2,58_ = 2.98, *p* = 0.09	F_1,58_ < 1	F_2,58_ < 1
Time in the upper third, %	F_2,58_ = 2.42, *p* = 0.1	F_1,58_ < 1	F_2,58_ < 1

Statistically significant values are marked in bold.

**Table 3 biomolecules-11-01421-t003:** Factor loadings (correlation coefficients) for travelled distance, immobility time, distance from the tank’s bottom, explored part of the tank, time spent in the lower and upper thirds of the tank in two-, four- and six-month-old males of *N. furzeri.*

Trait	Factor 1 (51.9%)	Factor 2 (40.8%)
Distance traveled, cm	r = −0.10, *p* > 0.05	**r = 0.96, *p* < 0.0001**
Immobility time, %	r = 0.10, *p* > 0.05	**r = −0.98, *p* < 0.0001**
Explored part, %	r = −0.40, *p* > 0.05	**r = 0.78, *p* < 0.001**
Distance from the bottom, cm	**r = −0.99, *p* < 0.0001**	r = 0.05, *p* > 0.05
Time in the lower third, %	**r = 0.98, *p* < 0.0001**	r = 0.0, *p* > 0.05
Time in the upper third, %	**r = −0.97, *p* < 0.0001**	r = 0.0, *p* > 0.05

Statistically significant values are marked in bold.

**Table 4 biomolecules-11-01421-t004:** Two-way ANOVA of the effect of “Age”, “Sex” factors and their interaction on the variability of mRNA level of *Polr2e*, *Tph1a*, *Tph1b*, *Tph2*, *Slc6a4b*, *Mao*, *Htr1aa*, *Htr1b*, *Htr1f*, *Htr2a*, *Htr2Cl1*, *Htr3a*, *Htr3b*, *Htr4*, *Htr5ab*, *Htr6*, and *Htr7* genes in the brain of *N. furzeri*.

Gene	Age	Sex	Interaction
*Polr2eb*	F_2,54_ < 1	F_1,54_ < 1	F_2,54_ < 1
*Tph1a*	F_2,54_ < 1	**F_1,54_ = 17.38, *p* < 0.001**	F_2,54_ < 1
*Tph1b*	**F_2,54_ = 3.51, *p* = 0.037**	**F_1,54_ = 24.49, *p* < 0.001**	**F_2,54_ = 4.17, *p* = 0.021**
*Tph2*	F_2,54_ = 2.62, *p* = 0.082	**F_1,54_ = 12.45, *p* < 0.001**	F_2,54_ < 1
*Mao*	**F_2,54_ = 9.72, *p* < 0.001**	F_1,54_ = 1.54, *p* = 0.219	F_2,54_ < 1
*Slc6a4b*	**F_2,54_ = 3.41, *p* = 0.04**	**F_1,54_ = 23.57, *p* < 0.001**	**F_2,54_ = 5.15, *p* = 0.009**
*Htr1aa*	**F_2,54_ = 15.37, *p* < 0.001**	**F_1,54_ = 5.65, *p* = 0.021**	**F_2,54_ = 6.83, *p* = 0.002**
*Htr1b*	F_2,54_ = 2.38, *p* = 0.101	F_1,54_ < 1	F_2,54_ = 1.25, *p* = 0.295
*Htr1f*	F_2,54_ < 1	F_1,54_ < 1	F_2,54_ < 1
*Htr2a*	F_2,54_ = 2.98, *p* < 0.059	**F_1,54_ = 5.09, *p* = 0.028**	**F_2,54_ = 4.04, *p* = 0.023**
*Htr2lC1*	F_2,54_ = 2.15, *p* = 0.126	F_1,54_ < 1	F_2,54_ = 1.81, *p* = 0.174
*Htr3a*	**F_2,54_ = 3.54, *p* = 0.036**	**F_1,54_ = 20.36, *p* < 0.001**	**F_2,54_ = 4.46, *p* = 0.016**
*Htr3b*	**F_2,54_ = 4.83, *p* = 0.012**	**F_1,54_ = 23.12, *p* < 0.001**	**F_2,54_ = 4.37, *p* = 0.017**
*Htr4*	F_2,54_ = 2.93, *p* = 0.062	**F_1,54_ = 10.58, *p* = 0.002**	**F_2,54_ = 3.32, *p* = 0.044**
*Htr5ab*	F_2,54_ = 1.56, *p* = 0.219	F_1,54_ = 1.42, *p* = 0.239	F_2,54_ = 2.89, *p* = 0.064
*Htr6*	**F_2,54_ = 5.11, *p* = 0.009**	**F_1,54_ = 14.62, *p* < 0.001**	F_2,54_ = 1.86, *p* = 0.166
*Htr7*	F_2,54_ = 2.25, *p* = 0.116	F_1,54_ < 1	F_2,54_ < 1

Statistically significant values are marked in bold.

**Table 5 biomolecules-11-01421-t005:** mRNA level of Tph1a, Tph1b, Tph2, Slc6a4b, Mao, Htr1aa, Htr1b, Htr1f, Htr2a, Htr2cl1, Htr3a, Htr3b, Htr4, Htr5ab, Htr6, and Htr7 genes in the brains of two-, four- and six-month-old males and females of *N. furzeri*.

Age	2 Months of Age		4 Months of Age		6 Months of Age	
Sex	Male (11)	Female(8)	Male (12)	Female (9)	Male (12)	Female (8)
*Tph1a*	102.3 ± 3.4	80.9 ± 12 ^#^	96.2 ± 6.8	74.9 ± 5.3 ^#^	106.0 ± 7.5	73.0 ± 8.3 ^##^
*Tph1b*	11.1 ± 1.8	2.7 ± 0.5 ^#^	21.6 ± 3.0 **	3.1 ± 0.5 ^###^	8.4 ± 2.6 ^$$$^	4.1 ± 1.1
*Tph2*	15.4 ± 1.6	14.0 ± 0.9	24.0 ± 2.4 ***	11.2 ± 0.7 ^###^	16.6 ± 1.9 ^$$^	11.0 ± 0.7 ^#^
*Mao*	112.0 ± 5.2	106.4 ± 4.1	129.3 ± 5.1 *	125.0 ± 4.8 ^&^	135.5 ± 5.2 **	128.9 ± 7.1 ^&&^
*Slc6a4b*	12.6 ± 1.5	5.4 ± 0.6 ^#^	22.5 ± 3.6 **	5.7 ± 0.5 ^###^	10.6 ± 2.2 ^$$^	7.7 ± 1.3
*Htr1aa*	64.3 ± 2.5	66.0 ± 2.2	68.8 ± 3.3	51.7 ± 2.3 ^##,&&^	50.2 ± 2.4 **^,$$^	49.4 ± 3.2 ^&&&^
*Htr1b*	27.5 ± 2.7	29.1 ± 2.4	35.6 ± 3.3	30.0 ± 1.6	27.7 ± 1.6	28.4 ± 1.9
*Htr1f*	77.6 ± 7.0	81.0 ± 10.5	86.4 ± 7.3	72.0 ± 3.9	79.4 ± 3.8	74.0 ± 7.9
*Htr2a*	62.7 ± 2.1	75.3 ± 3.9 ^##^	77.3 ± 2.1 ***	72.9 ± 3.6	63.2 ± 2.4 ^$$$^	72.8 ± 5.7 ^#^
*Htr2cl1*	81.7 ± 3.5	87.3 ± 5.9	88.9 ± 4.9	78.8 ± 3.4	76.2 ± 3.1	76.8 ± 4.1
*Htr3a*	21.3 ± 2.7	13.9 ± 1.6	44.0 ± 6.7 ***	12.6 ± 0.9 ^###^	24.1 ± 5.0 ^$$$^	13.7 ± 1.4 ^###^
*Htr3b*	13.5 ± 2.3	4.3 ± 0.9 ^#^	26.7 ± 4.8 **	4.9 ± 0.8 ^###^	9.3 ± 2.8 ^$$$^	4.6 ± 0.7
*Htr4*	35.1 ± 1.7	32.7 ± 1.9	39.2 ± 3.4	25.2 ± 2.7 ^###^	29.7 ± 2.1 ^$$^	26.2 ± 1.5
*Htr5ab*	45.3 ± 1.3	50.1 ± 1.8	55.3 ± 3.4	46.1 ± 2.1	46.8 ± 2.5	45.6 ± 2.8
*Htr6*	15.0 ± 1.8	9.0 ± 1.6	26.9 ± 4.2 ***	13.0 ± 0.6 ^###^	16.3 ± 2.1 ^$$^	11.6 ± 1.9
*Htr7*	31.4 ± 3.4	33.1 ± 3.0	41.1 ± 3.5	35.2 ± 1.4	37.5 ± 2.1	35.2 ± 2.0

The data are presented as copies number of target gene/100 copies of *Polr2e* gene (internal standard). * *p* < 0.05, ** *p* < 0.01, *** *p* < 0.01 vs. two-month-old males; ^$$^
*p* < 0.01, ^$$$^
*p* < 0.001 vs. four-month-old males; ^#^
*p* < 0.05, ^##^
*p* < 0.01, ^###^
*p* < 0.001 vs. males of the same age; ^&^
*p* < 0.05, ^&&^
*p* < 0.01, ^&&&^
*p* < 0.01 vs. two-month-old females.

**Table 6 biomolecules-11-01421-t006:** Factor loadings for mRNA levels of Tph1a, Tph1b, Tph2, Slc6a4b, Mao, Htr1aa, Htr1b, Htr1f, Htr2a, Htr2cl1, Htr3b, Htr4, Htr5ab, Htr6, and Htr7 genes in the brain of two-, four- and six-month-old males of *N. furzeri*.

Gene	Factor 1 (56.3%)	Factor 2 (17.3%)	Factor 3 (8.9%)
*Tph1a*	r = 0.053, *p* > 0.05	r = 0.051, *p* > 0.05	**r = 0.91, *p* < 0.0001**
*Tph1b*	**r = 0.95, *p* < 0.0001**	r = 0.20, *p* > 0.05	r = 0.07, *p* > 0.05
*Tph2*	**r = 0.85, *p* < 0.0001**	r = 0.39, *p* > 0.05	r = 0.06, *p* > 0.05
*Mao*	r = 0.11, *p* > 0.05	**r = 0.93, *p* < 0.0001**	r = 0.06, *p* > 0.05
*Slc6a4b*	**r = 0.93, *p* < 0.0001**	r = 0.23, *p* > 0.05	r = 0.02, *p* > 0.05
*Htr1aa*	**r = 0.83, *p* < 0.0001**	r = 0.02, *p* > 0.05	r = −0.14, *p* > 0.05
*Htr1b*	**r = 0.64, *p* < 0.01**	r = 0.32, *p* > 0.05	r = 0.35, *p* > 0.05
*Htr1f*	**r = 0.65, *p* < 0.01**	r = 0.39, *p* > 0.05	r = 0.41, *p* > 0.05
*Htr2a*	r = 0.53, *p* > 0.05	r = 0.54, *p* > 0.05	r = −0.36, *p* > 0.05
*Htr2cl1*	**r = 0.82, *p* < 0.0001**	r = 0.34, *p* > 0.05	r = 0.15, *p* > 0.05
*Htr3a*	**r = 0.88, *p* < 0.0001**	r = 0.35, *p* > 0.05	r = 0.10, *p* > 0.05
*Htr3b*	**r = 0.90, *p* < 0.0001**	r = 0.12, *p* > 0.05	r = 0.09, *p* > 0.05
*Htr4*	**r = 0.87, *p* < 0.0001**	r = 0.26, *p* > 0.05	r = 0.09, *p* > 0.05
*Htr5ab*	**r = 0.81, *p* < 0.0001**	r = 0.43, *p* > 0.05	r = 0.05, *p* > 0.05
*Htr6*	**r = 0.86, *p* < 0.0001**	r = 0.41, *p* > 0.05	r = 0.13, *p* > 0.05
*Htr7*	r = 0.53, *p* > 0.05	**r = 0.64, *p* < 0.01**	r = 0.23, *p* > 0.05

Statistically significant values are marked in bold.

**Table 7 biomolecules-11-01421-t007:** Correlation between behavioral traits in the novel tank diving test and mRNA levels of 5-HT-related genes in the brain of two-, four- and six-month-old males of *N. furzeri*.

Genes	Travelled Distance	Immobility Time	Explored Part of the Tank	Distance from the Bottom	Time in the Lower Third	Time in the Upper Third
*Tph1a*	−0.11 (0.99)	0.04 (0.99)	−0.13 (0.99)	−0.18 (0.99)	0.22 (0.99)	−0.13 (0.99)
*Tph1b*	−0.27 (0.99)	0.26 (0.99)	−0.17 (0.99)	0.11 (0.99)	−0.13 (0.99)	0.11 (0.99)
*Tph2*	−0.30 (0.99)	0.28 (0.99)	−0.24 (0.99)	0.08 (0.99)	−0.13 (0.99)	0.09 (0.99)
*Mao*	−0.03 (0.99)	−0.05 (0.99)	−0.01 (0.99)	−0.04 (0.99)	0.02 (0.99)	−0.02 (0.99)
*Slc6a4b*	−0.29 (0.99)	0.27 (0.99)	−0.18 (0.99)	0.08 (0.99)	−0.10 (0.99)	0.09 (0.99)
*Htr1aa*	−0.44 (0.75)	0.43 (0.78)	−0.19 (0.99)	0.28 (0.99)	−0.29 (0.99)	0.25 (0.99)
*Htr1b*	−0.26 (0.99)	0.25 (0.99)	−0.32 (0.99)	−0.16 (0.99)	0.13 (0.99)	−0.19 (0.99)
*Htr1f*	−0.23 (0.99)	0.20 (0.99)	−0.09 (0.99)	0.02 (0.99)	−0.02 (0.99)	0.03 (0.99)
*Htr2a*	−0.40 (0.93)	0.38 (0.96)	−0.16 (0.99)	0.30 (0.99)	−0.33 (0.99)	0.23 (0.99)
*Htr2cl1*	−0.30 (0.99)	0.29 (0.99)	−0.29 (0.99)	0.01 (0.99)	−0.03 (0.99)	0.04 (0.99)
*Htr3a*	−0.24 (0.99)	0.21 (0.99)	−0.15 (0.99)	0.07 (0.99)	−0.10 (0.99)	0.07 (0.99)
*Htr3b*	−0.40 (0.93)	0.38 (0.96)	−0.30 (0.99)	0.11 (0.99)	−0.14 (0.99)	0.08 (0.99)
*Htr4*	−0.26 (0.99)	0.22 (0.99)	−0.02 (0.99)	0.31 (0.99)	−0.30 (0.99)	0.33 (0.99)
*Htr5ab*	−0.36 (0.98)	0.34 (0.99)	−0.14 (0.99)	0.19 (0.99)	−0.21 (0.99)	0.24 (0.99)
*Htr6*	−0.26 (0.99)	0.23 (0.99)	−0.14 (0.99)	0.11 (0.99)	−0.13 (0.99)	0.10 (0.99)
*Htr7*	−0.23 (0.99)	0.18 (0.99)	−0.20 (0.99)	0.03 (0.99)	−0.05 (0.99)	0.06 (0.99)

*p*-values (false discovery rate controlled by Bonferroni) are shown in brackets.

**Table 8 biomolecules-11-01421-t008:** Location of 5-HT related genes in the genome of *N. furzeri.*

*N. furzeri* Gene	Location (Ensemble)	Mammalian Homologue
*Tph1a*	sgr13: 18,976,907–18,981,171	TPH1
*Tph1b*	sgr07: 45,615,830–45,638,187	TPH1
*Tph2*	sgr13: 36,218,596–36,239,915	TPH2
*Mao*	sgr14: 7,478,504–7,513,691	MAOA, MAOB
*Slc6a4b*	sgr18: 22,023,293–22,030,954	5-HT transporter
* Htr1aa *	sgr18: 29,466,111–29,467,391	5-HT1A
*Htr1b*	sgr04: 71,348,837–71,349,973	5-HT1B
*Htr1f*	sgr14: 3,149,799–3,151,013	5-HT1F
*Htr2a*	sgr14: 12,123,202–12,202,559	5-HT2A
*Htr2cl1*	sgr04: 14,685,800–14,729,671	5-HT2C
*Htr3a*	sgr11: 48,000,514–48,007,311	5-HT3A
*Htr3b*	sgr11: 48,010,172–48,020,735	5-HT3B
*Htr4*	sgr01: 53,960,781–54,173,488	5-HT4
*Htr5ab*	sgr08: 18,341,119–18,361,475	5-HT5B
*Htr6*	sgr15: 33,021,207–33,028,620	5-HT6
*Htr7*	sgr18: 7,942,262–7,973,622	5-HT7

**Table 9 biomolecules-11-01421-t009:** Age-related alterations in tryptophan hydroxylases (TPH), monoamine oxidize (MAO), 5-HT transporter (SERT) and 5-HT receptors in the brains of *N. furzeri* and mammals.

Molecule	*N. furzeri* (mRNA)	Mammals (Protein, mRNA)
TPH	*Tph1a*: **0** m, f *Tph1b*: ↓ m, **0** f *Tph2*: ↓ m, **0** f	Tph2 mRNA: **0** rat [53]; TPH2 activity: ↑ midbrain ↓ medulla rat [54]
MAO	*Mao*: ↑ m, f	MAOA activity: ↑ rat [55], ↑ human [56,57,58]
SERT	*Slc6a4b*: ↓ m, **0** f	SETR protein: ↓ mouse [59], ↓ rat [60], ↓ hamster [61], ↓ human [62]
5-HT1A	*Htr1aa*: ↓ m, f	5-HT1A protein: ↓ rat [63], ↓ hamsters [61], ↓ monkey [64], ↓ human [65,66]; *Htr1a* mRNA: **0** rat [53,62]
5-HT1B	*Htr1b*: **0** m, f	5-HT1B protein: ↓ rat [69]; Htr1b mRNA: ↓ rat [53]
5-HT1F	*Htr1f*: **0** m, f	No data
5-HT2A	*Htr2a*: ↓ m, **0** f	5-HT2A protein: ↓ rats [64], ↓ monkey [64], ↓ human [66,69]; *Htr2a* mRNA: **0** rat [67]
5-HT2C	*Htr2cl1*: **0** m, f	*Htr2c* mRNA: **0** rat [67]
5-HT3	*Htr3a*: ↓ m, **0** f *Htr3b*: ↓ m, **0** f	No data
5-HT4	*Htr4*: ↓ m, **0** f	5-HT4 protein: ↓ human [70]
5-HT5	*Htr5ab*: **0** m, f	No data
5-HT6	*Htr6*: ↓ m, **0** f	5-HT6 protein: ↓ human [69,71]
5-HT7	*Htr7*: **0** m, f	*Htr7* mRNA: **0** hamster [72], **0** rat [67]

**0** no alteration, ↑ increase, ↓ decrease in old individuals. M—males, f—females.

## Data Availability

Not applicable.

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
