# Peer review of "Age-Related Alterations in the Behavior and Serotonin-Related Gene mRNA Levels in the Brain of Males and Females of Short-Lived Turquoise Killifish (*Nothobranchius furzeri*)"

_biomolecules, 2021, doi:10.3390/biom11101421_

Round 1

Reviewer 1 Report

In submitted manuscript the age dependent alterations in the 5- HT dependent behavior and mRNA levels of the 5-HT-related genes in the brain of short-lived turquoise killifish, Notobranchius furzeri have been examined. The authors studied the locomotor activity and anxiety-related behavior in the novel tank diving test as well as mRNA levels of all sequenced 5-HT-related genes such as Tph1a, Tph1b, Tph2, Slc6a4b, Mao, Htr1aa, Htr1b, Htr1f, Htr2a, Htr2cl1, Htr3a, Htr3b, Htr4, Htr5ab, Htr6, and Htr7 in brain of 2-, 4- and 6-month-old males and females of N. furzeri. Significant effect of sex on body mass as well as on mRNA level of Tph1a, Tph1b, Tph2, Slc6a4b, Htr1aa, 5-HT2a, Htr3a, Htr3b, Htr4, Htr6 genes was revealed. The marked effect of age on body mass, locomotor activity and the mRNA level of Tph1b, Tph2, Slc6a4b, Mao, Htr1aa, Htr2a, Htr3a, Htr3b, Htr4, Htr6 genes in brain of N. furzeri males was shown. Locomotor activity and expression of Mao gene increased, while expression of Tph1b, Tph2, Slc6a4b, Htr1aa, Htr2a, Htr3a, Htr3b, Htr4, Htr6 genes decreased in 6-month-old N. furzeri.

This work is the first study on the effects of sex and age of the brain 5-HT system and 5-HT- dependent behavior in the novel tank diving test in short-lived turquoise killifish.

The observed age-dependent alterations in 5-HT related genes in brain of N. furzeri reflect the fundamental changes in the vertebrate 5-HT system during aging. The results obtained by authors showed that N. furzeri is a perspective model to study the age-related alterations in the brain 5-HT system.

The manuscript contains new original data that are of great importance for the study of the

fundamental problems of age-related alterations in the brain 5-HT system.

Presented work is written in clear style, broadly covers key literature and represents a new advancement in neuroscience. The manuscript is carefully prepared and contains very good documentation. In my opinion the paper by Evsiukova1 et al. can be published in Biomolecules in the presented form.

Author Response

Thank you for your kind appreciation of our manuscript.

Reviewer 2 Report

Evsiukova and colleagues present an interesting study characterizing cross-sectional serotonin gene dynamics in brain tissue and anxiety-like behavior in killifish at three ages. This work adds important information to the field in terms of this rapid aging model's utility in understanding emotion and brain molecular biology and neurochemistry across the life course. However, there are some issues with the manuscript in its current form that should be addressed. They are as follows:

Major issues: 

Justification for measuring serotonin related genes rather than serotonin itself and its turnover (metabolism) is missing. 

In the PCA analysis, six variables explained the majority of the variance. However, since some of the variables dependent on one another I am not convinced that this much variance is really truly explained by these variables/factors. For instance, in Table 3, two of the three factor 1 variables are time in lower and upper regions of the arena. These are not independent states since if the fish is in the upper third it cannot be in the lower third and the distance to the bottom is redundant in some ways with the 1/3rd arena demarcations. Thus, I do not think PCA is necessary or informative for these types of discreet study designs with age and sex and the main factors resolved with means comparisons. In fact, I think this brings up the issue that there are perhaps too many variables collected in this novel tank diving assay. 

Related to the above, why not look at covariance of brain gene expression and behavioral readouts in the fish since the data are collected in a within-subjects manner? How does variability within age groups in 5-HT gene expression correlate with anxiety like behavior in the novel tank assay? And do any of those relationships break down with aging?

There is not acknowledgement that using the whole brain to measure gene expression could lead to Type II errors given that regions of the brain might show age or sex-dependent differences in expression that become washed out because of opposing effects in other brain regions when the whole brain is homogenized and assessed collectively. I am not suggesting that the authors need to look at this region by region. But the limitations of using the whole brain homogenates for expression profiling need to be acknowledged. 

The statement in line 377 that the reduced expression of certain genes can be attributed to neural degeneration is not supported by the data in the study as it was not measured. Please remove or change the writing to indicate that there is the potential that reduced expression is attributable to degeneration. 

Minor issues:

The sentence beginning "Six-eight month-old..." in line 43: was this meant to say "Six to eight month old N. furzeri"?

Section 2.2:  please list the duration of time fish are recorded in the test tank.

The sentence beginning "Therefore,..." on line 414 needs to be rewritten for clarity. Perhaps something like: "Therefore N furzeri is a promising model that can significantly..." Similarly, I suggest that the word "object" be replaced by "model" the first sentence of the abstract (line 12).

Author Response

  1. We added justification for measuring serotonin related genes in lines 58-61.
  2. In our study we used classical for novel tank diving test behavioral traits [22-24,36,37]. However, you’re right and some of these traits can correlate with each other. We used PCA in order to revealed the groups of traits linked by correlations. Moreover, PCR provides a compact view of age-related alterations.
  3. We added the analysis of correlations between the behavioral traits and the genes expression in chapter 3.4 (lines 272-278), Table 7 and in the fin of Discussion section (lines 421-426).
  4. Unfortunately, due to methodical limitations we cannot reliably measure mRNA level in small regions of killifish brain. This may be a cause of our failure to show correlation between the age-related alterations in the behavioral traits and the gene expression (lines 421-426). Nevertheless, we revealed sex- and age-related alterations in some genes expression at the level of whole killifish brain (lines 439-441).
  5. We removed the speculation that the reduced expression of certain genes can be attributed to neural degeneration, since it did not supported experimental evidences.
  6. We replaced "Six-eight month-old..." in line 42 by "Six to eight month old…”
  7. We added the recording duration in line 97.
  8. We replaced “Therefore, N. furzeri is a quite perspective model object…” by “Therefore, N. furzeri is a promising model species…” (line 460). We also replaced ‘object’ by ‘model organism’ in the Abstract (line 12).

Thank you very much for your friendly and valuable comments.

Reviewer 3 Report

See the PDF attachment for comments.

Author Response

  1. We replaced “laboratory object” in line 12 by “model organism”.
  2. We replaced “For the first time in the present paper…” in line 13 by “In the present paper we study for the first time...”
  3. We replaced the list of genes in the Abstract by “various 5-HT-related genes@ (line 14”.
  4. We put the age effect before sex effect in the Abstract (lines 15-21).
  5. We replaced “N. furzeri is a suitable model to study the fundamental problems of age-related alterations in the brain 5-HT system” by “N. furzeri is a suitable model to study the fundamental problems of age-related alterations in various mRNA levels related with the brain 5-HT system” (lines 21-22).
  6. We removed keywords “behavior” and “mRNA” and added keywords “novel tank diving test” and “gene expression” (lines 23-24).
  7. We added “various” before “kinds of behavior” (line 28).
  8. We added “the” before “relatively” and replaced “life-span” by “life span” (line 36).
  9. We replaced “Notobranchius” by “Nothobranchius” (line 38).
  10. We added two publications [18,19] concerning the effect of fluoxetine on killifish behavior in the Introduction (lines 46-48) and in the list of references.
  11. We added the phrase “However, until now there is only one publication revealing no effect of sex and rearing conditions the level and turnover of 5-HT in brain of 108-day-old N. furzeri [20]” in the Introduction (lines 48-50).

10-11. We changed “for elucidation of the role that the brain 5-HT system plays in aging” by “to elucidate the alterations in the brain 5-HT system and 5-HT-related functions in aging” (lines 52-53).

  1. We added “the” before “ZMZ1001 strain” (lines 68-69).
  2. We replaced “mix groups” by “mixed-sex groups” (line 75).
  3. We replaced “144 liters” by “125 liters” (line 75).
  4. In our facility constant temperature 27C is kept. However, it can vary 1C when water is substituted by tap water (lines 78 and 82).
  5. We changed “ tank was” by “tanks were” (line 79).
  6. We use plastic plants a environmental enrichment and as shelters from aggressive conspecifics (lines 80-81).
  7. We replaced “ad libitum” by “until satiation” (line 84).
  8. This problem has two solution: use different tanks for different age groups and take fish of different age from the same tank. Both solution are not ideal. We chose the second solution in the hope that the time-dependent alterations in fish density do not markedly affect the behavior and the brain 5-HT system in fish (lines 76-77).
  9. The recordibf starts immediately after placing fish in the test tank (line 95).
  10. We nevertheless decided to leave the technical details about the Web camera and its connection.
  11. We added “in addition” before “the number of variable…” (line 141) and specified the Statistic software (line 142).
    24. F<1 means absence of statistically significant difference, however in this case p-value is meaningless (that is different from p>0.05). So, F<1, p>0.05 is not correct from formal mathematics.
  12. We placed the age-effect before the sex-effect (lines 154-157).
  13. We prefer the present markers because they reflect p-values, while letters as markers indicate only difference and do not reflect p-value.
  14. For PCA factors comparison we used discriminant analysis (not ANOVA) which also uses F-statistics.
  15. We added the information about biological content of PCA factors in the legends of Figures 3 and 4.
  16. We made in bold all column titles in Table 5.
  17. We added in the first phrase of the Discussion that we studied the behavior in novel tank diving test and the levels of mRNA of 5-HT related genes in the brain in three age groups of killifish (lines 284-285).

31-32. We replaced “It” by “This” , “confirmes” by “confirms” and add more references concerning sexual dimorphism in body size and mass (line 288).

33-35. We removed the phrase those repeat the results and discussed our results in the light of published information about behavior of killifish (lines 306-325).

  1. We removed phrase “Unfortunately, the possible role of the brain 5-HT 294 system in the regulation of N. furzeri behavior is still obscure” and added more information about the 5-HT involvement in the regulation of killifish’s behavior (lines 329-333).
  2. The term “reuptake” is widely used in 5-HT-related literature. We added “the” before “synaptic” (line 347).
  3. We removed “gene” before “Scl6a4b” (line 350).
  4. We changed “were” to “was” and replaced “well” after “agrees” (line 377).
  5. We changed “dynamics” to “dynamic” (lines 379-382).
  6. We replaced “with that in mammalian brain” by “with that of mammalian brain” (line 389).
  7. We added “and” before “humans” in line 393.

43-44. We added “the” before “brain” in lines 395 and 402.

  1. We made this phrase less bold (lines 407-408).
  2. We charged “It” to “This” in the firs sentence of the Conclusion (line 428).

47-48. We replaced “model object” by “model species” (line 460) and “accelerates” by “accelerate” (line 461).

Thank you very much for your friendly and valuable comments

Round 2

Reviewer 3 Report

I commend the authors on their effort to improve the manuscript. All my comments were considered, and the authors either amended the manuscript accordingly or provided sufficient argumentation for why they didn't follow my suggestions. The manuscript is now more nuanced and provides better context. Personally I think the varying size of the social groups could be an important confounding factor with relation to behavior and 5-HT related gene expression, but this is now disclosed in the M&M section.

The manuscript needs some small grammatical revision still to cross the i’s and cross the t’s. Below, I identified some points that need correcting – this can be done during the proofreading stage.

Line 14: Add ‘the’ before ‘brain’.

Line 16: Add ‘the’ before ‘brain’.

Line 17: Add ‘the’ before ‘Mao gene’.

Line 19: Add ‘A’ before ‘Significant’.

Line 48: ‘N. furzeri’ in italics.

Line 50: Add ‘on’ before ‘the level’.

Line 95: Change ‘immediarely’ to ‘immediately’.

Line 121: Add ‘the’ before ‘small’.

Line 284: Add ‘the’ before ‘novel’.

Line 311: Add ‘the’ before ‘present’.

Line 315: ‘N. furzeri’ in italics.

Line 315: Put a comma after ‘study’.

Line 424: Add ‘the’ before ‘novel diving test’.

Line 425: ‘could correlated’ probably should be ‘correlated’?

Line 440: Change ‘could revealed’ to ‘revealed’.

Line 460: Remove the comma after ‘species’.